# The Relationship of the Mechanisms of the Pathogenesis of Multiple Sclerosis and the Expression of Endogenous Retroviruses

**DOI:** 10.3390/biology9120464

**Published:** 2020-12-11

**Authors:** Vera R. Lezhnyova, Ekaterina V. Martynova, Timur I. Khaiboullin, Richard A. Urbanowicz, Svetlana F. Khaiboullina, Albert A. Rizvanov

**Affiliations:** 1Institute of Fundamental Medicine and Biology, Kazan Federal University, 420008 Kazan, Russia; veralezhnyova@gmail.com (V.R.L.); sv.khaiboullina@gmail.com (S.F.K.); rizvanov@gmail.com (A.A.R.); 2Republican Clinical Neurological Center, Republic of Tatarstan, 420021 Kazan, Russia; timuur@gmail.com; 3Wolfson Centre for Global Virus Infections, University of Nottingham, Nottingham NG7 2RD, UK; richard.urbanowicz@nottingham.ac.uk; 4School of Life Sciences, University of Nottingham, Nottingham NG7 2UH, UK; 5Department of Microbiology and Immunology, University of Nevada, Reno, NV 89557, USA

**Keywords:** HERV-W, ERVWE1, MSRV, multiple sclerosis, citrullination, CpG methylation

## Abstract

**Simple Summary:**

Multiple sclerosis is a neurodegenerative disease of the central nervous system, develops at an early age and often leads to a disability. The etiological cause of the disease has not been fully elucidated, and as a result, no effective treatment is available. This review summarizes the current knowledge about the relationship between the expression of human endogenous retroviruses and the pathogenesis of multiple sclerosis. The epigenetic mechanisms of transcriptional regulation, the role of transcription factors, cytokines, and exogenous viruses are also addressed in this review. The elucidation of the mechanisms of an increase in endogenous retrovirus expression in multiple sclerosis could help to develop therapeutic strategies and novel methods for early diagnosis and treatment of the disease.

**Abstract:**

Two human endogenous retroviruses of the HERV-W family can act as cofactors triggering multiple sclerosis (MS): MS-associated retrovirus (MSRV) and ERVWE1. Endogenous retroviral elements are believed to have integrated in our ancestors’ DNA millions of years ago. Their involvement in the pathogenesis of various diseases, including neurodegenerative pathologies, has been demonstrated. Numerous studies have shown a correlation between the deterioration of patients’ health and increased expression of endogenous retroviruses. The exact causes and mechanisms of endogenous retroviruses activation remains unknown, which hampers development of therapeutics. In this review, we will summarize the main characteristics of human endogenous W retroviruses and describe the putative mechanisms of activation, including epigenetic mechanisms, humoral factors as well as the role of the exogenous viral infections.

## 1. Endogenous and Exogenous Factors Affecting the Development of MS

In recent years, the incidence and prevalence rates of multiple sclerosis (MS), a neurodegenerative disease, have increased worldwide. MS is a chronic, heterogeneous, immune-associated disease characterized by the formation of loci of inflammation and damage of myelin sheath of nerve fibers in the brain and spinal cord, leading to a persistent disability [1]. There are multiple hypotheses proposed to establish the etiology of MS, showing the multifactorial nature of the disease. There are several risk factors associated with the disease including gender (often diagnosed in women) and age (most common in young age) of the patient [2,3]. Hormone levels have also been implicated in late onset of the disease, as the first episode of MS neurological symptoms is often diagnosed in older women who have been pregnant as compared to those who have not [4,5]. The hormones appear to contribute to decreased relapse rates during pregnancy and reoccurring after the delivery [6]. Numerous risk factors and the multifactorial etiology of the disease makes early diagnosis challenging [7]. This makes development of the etiotropic therapeutics particularly difficult.

Several endogenous and exogenous risk factors have been identified for this disease. One potential endogenous risk factor of the illness is vitamin D deficiency [8]. Vitamin D is synthesized in the skin, where ultraviolet radiation initiates the first non-enzymatic reaction of the synthesis of the precursor of the active form of vitamin D, calcitriol. Numerous studies have shown a correlation between a decrease in the level of ultraviolet radiation with an increase of the latitude of the region of residence and an increase in the risk of MS and relapse of disease [9,10,11]. The role of vitamin D in MS was further confirmed by Ramagopalan et al., as an increased binding of vitamin D receptor (VDR) was found on genes associated with risk of MS [12]. Another endogenous factor contributing to the risk of MS is believed to be immune auto-aggression against self-proteins. It has been shown that cytotoxic T-lymphocytes are involved in the formation of loci of inflammation and demyelination, and are also associated with the severity of MS [13]. In vitro experiments have shown a potential connection between vitamin D and immune responses in MS. It has been demonstrated that human dendritic cells, in the presence of calcitriol, increase expression of CD31, which can reduce their ability to prime CD4+ T lymphocytes [14]. Low levels of vitamin D can disrupt this regulation on CD4+ T lymphocyte activity, leading to an increased production of pro-inflammatory cytokines, often found in MS [15]. The activity of the TAGAP gene (T-Cell Activation RhoGTPase Activating Protein), which is responsible for the Cdc42-dependent maturation of CD4 + T-lymphocytes [16], can be suppressed by direct exposure to calcitriol [17], suggesting an additional mechanism of vitamin D contribution to pathogenesis of MS.

Genetic predisposition and accumulation of mutations in genes associated with MS could play an essential role in pathogenesis of the disease. GWAS (genome-wide association studies) data have shown a missense mutation in the CD226 gene in MS leads to an amino acid change in the DNAM-1 receptor, thereby blocking or changing its function [18]. This receptor is responsible for NK cells’ cytolytic activity upon interaction with the poliovirus receptor (PVR) on the surface of activated CD4+ T lymphocytes in MS [19]. The single-nucleotide polymorphism (rs243324) effects expression in the 5′-regulatory region of the SOCS1 gene [20], which is involved in regulating cytokine signals, and has impaired expression in patients with MS [21].

Exogenous factors also include viral triggers of MS. There are many hypotheses that explain the possible mechanism of MS caused by viruses. HHV-4 (EBV) virus is one of those most commonly linked to MS, as was shown by the theory of molecular mimicry [22,23]. A higher frequency of cross-reactive T cell progenitors to myelin basic protein and HHV-6 in MS patients compared to controls has been shown [24]. In vivo experiments have demonstrated that stimulation of CD4 + CD28null T cells by human cytomegalovirus (CMV), another member of family *Herpesviridae*, causes them to proliferate, which can initiate migration of lymphocytes and damage to the central nervous system [25].

Other exogenous factors of MS could be bacterial infections, as microbial PAMPs can activate antigen presenting cells (APCs) and subsequently cause proliferation of autoreactive T and B lymphocytes. Similar to the molecular mimicry seen with HHV-4, the epitopes of some bacterial peptides are similar to host antigens and serve to form autoreactive T-cells [26]. Evidence of this was the detection of a significant immune response to the protein Mycobacterium avium subsp. paratuberculosis, which has homology with the C region of the gamma chain of the T cell receptor, in MS [27]. Some studies show a correlation between the incidence of MS and the presence of antibodies to C. pneumoniae, but the exact role of these antibodies in pathogenesis of the disease remains unclear [28,29]. In vivo studies have shown that microbiota from MS twins transplanted to transgenic mice with the T cell receptor specific for myelin oligodendrocyte glycoprotein (MOG), caused a higher level of autoimmunity than the microbiota obtained from healthy twins [30]. These data provide strong evidence on the role of bacteria in the pathogenesis of MS.

Many studies have shown the relationship between various neurodegenerative diseases (MS, amyotrophic lateral sclerosis, Alzheimer’s disease) and expression of endogenous retroviruses genes [31,32]. The expression of endogenous retroviruses can be detected not only in healthy people, for example during the pregnancy [33], but also in various pathologies [34]. Expression of the HERV-W (MS-associated retrovirus env) protein was found in the brain tissue of all studied MS patients during active demyelination at an early stage and during late disease progression [35]. Additionally, the HERV-Wenv protein was found in the serum, infiltrated macrophages, perivascular infiltrates and activated microglial cells of patients with various forms of MS [36]. In a study by Kremer et al., it was found that microglial phagocytosis was inhibited by the HERV-Wenv protein, which in turn leads to the accumulation of myelin debris, which prevents neuronal remyelination [37]. Additionally, this protein reduces the expression of neuroprotective molecules by microglial cells [37]. It is worth noting that HERV-W Env epitopes were discovered on the surface of B cells and monocytes in samples of patients with the active and stable relapsing–remitting form of the disease [38]. The fact that the HERV-Wenv protein is found in the brain tissues, as well as in the serum and cerebrospinal fluid [39] of patients with MS, suggests that it is in volved in the pathogenesis of the disease and should be carefully studied.

## 2. General Characteristics of HERV-W

In the early 1970s, endogenous retroviral elements were discovered, which, as was shown later, were attained by our ancestors more than 25 million years ago and are constitutively present in the human genome [40,41,42]. These genetic elements were attained in human DNA as the result of the integration of exogenous retroviral RNA followed the reverse transcription [43]. It was suggested that the expression of these viral proteins have had a significant impact on human evolution [32,44]. For example, the glycosylated product of the *ERVWE1* gene, located on chromosome 7q21.2 [45], is the fusogenic protein Syncytin-1 (or *enverin*), which is involved in the fusion of cell membranes during the formation of syncytiotrophoblast during embryogenesis [33].

Human endogenous retroviruses (HERVs) are present in the genome in the form of a provirus and are inherited by vertical gene transfer [46]. From the moment of integration into the human genome, endogenous retroviruses have accumulated many mutations. That is why almost all retroviral inserts currently do not have open reading frames (ORFs), which are necessary for virus replication [47]. All HERVs are grouped into three classes based on the similarity of their sequences with infectious retroviruses [48]. The HERV terminology is based on the type of tRNA that binds to the primer binding site (PBS) [49].

In the context of MS, the study of *HERV-W env* gene expression is of most interest, since its product can activate the innate immune response through interaction with the TLR-4 receptor [50]. This process contributes to the activation of nitrosative stress, leading to impaired differentiation of oligodendrocytes and impaired remyelination of neurons [51]. In an in vitro study, the expression of the MS-associated retrovirus (MRSV) recombinant protein in the brain microvascular endothelial cells increased the expression of ICAM-1, which is associated with the interaction of the MSRV protein with TLR-4 receptors on the surface of endothelial cells [52]. The HERV-W family has approximately 654 elements that are permanently present in the human genome [53]. In the assembly of the human genome version GRCh37/hg19, 135 pseudogenes and 65 HERV-W proviruses were detected [54]. ORFs for *HERV-W env* were found on Xq22.3 [55] and 7q21.2 [56], chromosomes encoding the incomplete HERV-W envelope protein and Syncytin-1, respectively. *HERV-W env* was found in macrophages infiltrating the brain parenchyma and in perivascular infiltrate in patients with MS [36]. The exact reasons for the increased expression of the HERV-W family genes in MS are currently unknown.

## 3. The Structure of the HERV-W Family and the *ERVWE1* Gene

The HERV-W consists of four genes: *gag*, *pro*, *pol*, *env* (Figure 1). The *gag* gene encodes the structural components of the matrix, capsid and nucleocapsid, while the *pro* gene encodes the protease, the *pol* gene encodes for the reverse transcriptase and integrase, and the *env* gene encodes the precursor of the envelope glycoprotein. At the 5′ and 3′ ends, these genes are flanked by two identical long terminal repeats (LTR) [57]. LTRs consist of a U3 region containing a primer binding site (PBS), which uses tRNA^Trp^ to synthesize negative DNA strands, CAAT, and TATA box binding sites for transcription factors [58]. Additionally, in the U3 region of the 5′LTR *ERVWE1*, there is a phosphorylated cAMP response element-binding protein (CREB) binding region that is active in all cell types, as well as Sp-1, AP-2 binding sites containing GC-rich domains, GATA, Oct-1 and PPAR-γ/RXR [59]. The repeat (R) sequences region of *ERVWE1* contains a CAP transcription initiation site at the 5′ end and a polyadenylation site at the 3′ end [58], and it also serves as the primary site for transcription initiation [59]. Recently, an AG-rich non-coding, 2kb *pre-gag* region between the 5′-LTR and the *ERVWE1 gag* gene, whose functions are currently unknown, has been discovered [54]. Additionally, upstream regulatory element (URE), relevant to *ERVWE1*, includes the trophoblast specific enhancer (TSE) containing AP-2, Sp-1, and GCMa binding sites and flanked by MaLR (mammalian apparent LTR retrotransposon). The cAMP/PKA pathway’s role has also been demonstrated in the regulation of LTR promoter activity in *HERV-W env* [59]. Interestingly, Schmitt et al. provided evidence of the promoter activity of incomplete LTR sequences [60]. It was also shown that the LTRs with a U3R region were capable of generating antisense transcripts, while LTRs with a U5R region can facilitate sense gene transcription [60].

## 4. Epigenetic Mechanisms of Regulation of HERV-W Expression

### 4.1. RNA Interference

Understanding the epigenetic mechanisms helps to establish the relationship between pathological conditions and changes in gene expression regulation. MicroRNAs (miRNAs), which are part of the RNA-induced silencing complex (RISC), can inhibit gene expression at the level of transcription by the RNA interfering mechanism [61] and translational level [62], by suppression of mobile genetic elements replication, including HERV [63]. miRNA can bind to the 3′-end of the untranslated region (UTR) of the target mRNA leading to mRNA cleavage or prevention of subsequent translation [64] (Figure 2).

Currently, there are contradictory data about miRNA effects on development of MS and HERV-W expression. A significant decrease in miRNA-17 and miRNA-20a levels was found in the whole blood from patients with primary progressive (PP), secondary progressive (SP), and relapsing–remitting MS (RRMS) [65]. It was shown that these miRNAs are involved in regulation of genes responsible for T cell activation in MS [65]. Additionally, treatment of RRMS with a monoclonal antibody, natalizumab, decreased miRNA-17 level in CD4+ cells during remission, while it increased during the relapse phase of the disease [66]. Interestingly, the use of natalizumab led to a gradual decrease in the level of *MSRV env* transcripts and syncytin-1 in peripheral blood mononuclear cells of MS [67]. These conflicting results could be explained by the fact that each study used different patient populations without considering the relapse phase of disease.

Nineteen miRNAs were found to have more than 80% homology to HERV-W [68]. Their expression correlates with the low replication of HERVs genes in healthy tissues [68]. Huang and colleagues found short interfering RNAs (siRNAs) with a high level of time-dependent knockdown effect on syncytin-1 mRNA in vitro [33]. It could be suggested that the mechanism of RNA interference could indirectly affect the expression of HERVs. This assumption is supported by the finding that siRNA to the p65 gene of the NF-κB subunit could reduce URE-LTR activation of the *ERVWE1* promoter by TNFα [69].

### 4.2. Citrullination

Citrullination is one of the mechanisms of post-translational modification of proteins catalyzed by proteins of the Peptidyl Arginine Deiminase (PAD) family. Heterochromatin protein 1 α (HP1α) recognizes methylated histone H3K9 accumulated on the *HERV-W/ERVWE1* LTR, which leads to transcription repression (Figure 3A). An elevated level of citrullinated histone H3cit8K9me3 has been detected in the *HERV-W/ERVWE1* promoter region in MS [70]. It has also been shown that an increased PADI4 expression could contribute to upregulation of *HERV-W*/*ERVWE1*, by converting the H3K9me3 histone to the citrullinated form H3cit8K9me3 (Figure 3B). Local increased citrullination of H3R8 histone, located upstream of H3K9, could inhibit the binding of HP1α to H3K9me3 histone, which has been shown to be associated with increased expression of *HERV-W/ERVWE1* [70].

Antibodies to the myelin basic protein (MBP) have been detected in the central nervous system of MS patients [71]. An increase in the level of citrullinated MBP has been detected in the white matter of the brain of MS patients [72,73], ranging from 45% to 80%, compared to 18% found in healthy controls [74,75]. Citrullination changes the conformation of MBP and thus contributes to the formation of a weaker calmodulin binding site [72], which can affect the Ca^2+^-dependent activation of the large-conductance Ca(2+)-activated K(+) (BKCa) channels involved in depolarization of the cell membrane [73]. These data suggest that the changes in citrullination of MBP could be used as a biomarker for diagnosis of MS as well as for evaluation of the disease progression.

Peptidylarginine deiminase (PADI)-2 and PADI4 proteins responsible for citrullination were found expressed on the myelin sheath, where PADI2 appears to be more citrullinated on the arginine residues as compared to PADI4 [76]. Each substitution of arginine with citrulline causes a change in the MBP charge, which makes it more difficult to interact with the phospholipid layer of the myelin sheath [76]. In MS, the amount of PADI2 and PADI4, and, therefore, citrullinated proteins, has been shown to be increased [76]. A larger amount of PADI2 is localized in the periaxonal regions, which could explain why this end of the oligodendrocyte is the first to undergo to apoptosis [76]. However, it appears that citrullination of MBP has a limited role in activating T lymphocytes, and, therefore, unlikely to play a role in the development of MS [77]. Therefore, the role of citrullination of MBP in the pathogenesis of MS remains unclear.

MS patients have a persistent visual impairment, which is mainly associated with bilateral internuclear ophthalmoplegia and bilateral visual neuropathy [78]. The increased intraocular pressure found in MS [79], may be due to an increased intracellular calcium level. Interestingly, it was shown that an increased intraocular pressure could lead to upregulation of PADI2 expression [80]. Elevated levels of PADI2 and protein deimination were shown to be associated with pathological changes in MS [81], suggesting a role of this protein in pathogenesis of the disease.

### 4.3. Methylation and Acetylation

DNA methylation at the 5th position of the pyrimidine ring within the cytosine CpG site is a mechanism for gene expression regulation. Hypermethylation of the CpG site in the coding region of the gene can enhance expression [82]. Additionally, hypermethylation in the promoter or enhancer region makes the region heterochromatic, preventing transcription activators to bind to it, which leads to silencing of that gene [83] (Figure 4). Thus, hypermethylation in the LTR region of HERV-W can potentially inhibition the expression of these genes. It has been demonstrated that inactive unmethylated CpG sites in the promoter region can correlate with an increased Lys4 dimethylation in H3 histone, which may indicate that this mechanism could protect DNA from methylation [84]. In MS, a decreased expression of the *EHMT2* gene encoding the histone H3 methyltransferase has been demonstrated [85]. Interestingly, methylation of this gene has been shown to be necessary for repression of HERV-W transcription [70]. An increased *EHMT2* expression is also found in the late stages of oligodendrocyte differentiation, and shown to contribute to H3K9 methylation and chromatin compaction necessary for differentiation of oligodendrocyte progenitor cells (OPCs) [86], which is impaired in patients with MS [87].

The role of DNA methylation transcription of HERVs has been extensively studied. HERVs’ retroelements that are expressed in many types of tumors are less methylated than those in healthy tissues [88]. Hypomethylation of LTRs has been demonstrated in placental tissues, normally expressing *ERVWE1*, while in other tissues this region is strongly methylated [89]. Weber et al. showed that the insertion of foreign DNA can affect the methylation of CpG sites in various regions of human genes and change their expression [90]. They later found that transfection of cells with the same foreign DNA did not affect methylation of CpG sites and the level of HERV-W transcription [91]. These results suggest that exogenous retroviral elements are not affecting the expression of endogenous retroviruses by using the CpG site methylation mechanism.

An increased level of expression of DNA methyltransferase-3 (DNMT3B) beta subunits has been detected in MS [92], as has a lower demethylase content in the demyelinated hippocampus compared to the myelinated [92]. As a result of demyelination, a decrease in the methylation state of genes responsible for neuronal death, coupled with an increase in the methylation of genes responsible for neuron survival was demonstrated [92]. It is known that CD8+ T lymphocytes can be locally reactivated at lesions in MS [93]. A study of the degree of methylation of CpG sites in CD4+ and CD8+ T lymphocytes revealed significant hypermethylation in the DNA of CD8+ T lymphocytes in MS, especially at the location upstream of the transcription start site (TSS) and the first exon of genes [94].

Histone acetylation is a mechanism for regulating gene expression. The acetylation of the lysine in the N-terminal residue removes the positive charge from the histone [95], which reduces the strength of its interaction with the negatively charged phosphate groups in the DNA. This results in the chromatin becoming less compact, which facilitates the transcription of genes [95]. Interestingly, treatment of cells with histone deacetylase inhibitors (HDACs) did not increase HERV expression; however, when panobinostat, a histone acetylase inhibitor, was used, expression of *HERV-W env* was suppressed [96].

## 5. Changes in LTR Regions and the Effects of Transcription Factors

LTRs are the main regulatory elements of endogenous human retroviruses [58]. Several mutations in the 3’-LTR region are associated with an increased expression of *ERVWE1* mRNA as well as protein [97]. For example, the 142T > C mutation was shown to promote the formation of an additional binding site with c-Myb in the 3′-LTR and increase its activity [97]. However, analysis of the expression of proto-oncogenes in the white matter of the MS brain did not find a significant increase in c-Myb expression [98], while expression of c-Fos increased [98]. The increased expression of the HERV-W loci is explained by stimulation of the U3 region initiated by the 5′R LTR region [99]. The number of HERV transcripts from the 3′ end were lower than from the 5′ end, which could be explained by the distance from the U3 initiation site of reverse transcription [57]. It should be noted that intron proviral elements with single LTRs showed a higher level of transcription than elements having both LTRs [57]. Therefore, the role of these regulatory regions in HERV transcription requires further study.

## 6. The Role of the Immune System in Activation of HERV-W Transcription

Recently, the role of cytokines in the activation of endogenous retroviruses was demonstrated in MS. It was found that cytokines often associated with MS, such as tumor necrosis factor α (TNFα) and interferon-γ (IFN-γ), activate the *ERVWE1* promoter [69]. The highest activation of HERV-W expression in glioblastoma cells was detected when exposed to TNFα [69]. The mechanism of transcription activation by this cytokine was shown as associated with an increased binding of p65 of the NF-κB subunit to a sensitive element located in the enhancer region of the *ERVWE1* promoter [69].

The permeability of the blood–brain barrier (BBB) is inextricably linked with the relapse in patients with RRMS [100]. The CD9 protein, a tetraspanin family member, the expression of which is increased in the loci of inflammation in MS brains, has been shown to be involved in BBB permeability [101]. Overexpression of CD9 increases interaction with the G protein, which activates the cAMP/PKA signaling pathway, and this in turn leads to activation of Glial Cells Missing Transcription Factor 1 (GCM1), which binds to the regulatory region of *ERVWE1* and enhances gene expression [102,103] (Figure 5).

## 7. Activation of HERV-W Transcription upon Exposure to Exogenous Viruses

The effect of virus infection on the activation of endogenous human virus transcription has been extensively investigated. Studies have shown that human immunodeficiency virus (HIV) is commonly associated with HERV-W expression [104]. It was found that HIV trans-activator of transcription (Tat) protein increases the level of *MSRVenv* and *HERV-Wenv* mRNA in blood cells and astrocytes [104]. In monocytes, Tat can stimulate *MSRVenv* and inhibit *ERVWE1*, yet when differentiated into macrophages, the expression of both retroviral elements has been demonstrated. In astrocytes, HIV indirectly activates transcription of *MSRV* and *ERVWE1* due to interaction with TLR4 and the induction of TNFα [104]. Therefore, these expression on retroviral elements and worsening MS symptoms could be explained by antiretroviral therapy in patients with HIV [105].

Human cytomegalovirus (HCMV) could also be implicated in activation of HERV-W expression. It was found that when HCMV enters the cell, the reverse transcriptase, required for transcription of HERV, appears to be activated [106]. In vitro experiments have demonstrated the effect of the pleiotropic trans-activator of genes (Infected Cell Polypeptide 0 (ICP0)), which is synthesized shortly after infection with herpes simplex virus 1 (HSV-1), on the transcriptional activation of both *HERV-Wenv* and *MSRVenv* genes [107]. Human Herpesvirus 6A (HHV-6A) can activate *MSRVenv* expression by interacting with the extracellular domains of SCR3 and SCR4 of the isoform of CD46-Cyt1 receptor phosphorylated by tyrosine protein kinase C [108]. However, none of the exogenous viruses have been identified as a sole etiological factor of MS.

## 8. Conclusions

The pathogenesis of MS is still the subject of debate among scientists and health care providers. Symptomatic and pathogenetic treatment used in MS does not completely abrogate the neurodegeneration and the progression of the disease. Currently, MS is characterized as a multifactorial disease, in which a key role can be attributed to the external and genetic factors. One of the unresolved questions of the disease pathogenesis is the lack of understanding of the mechanism of HERV-W expression in MS, especially in the inflammatory foci within the brain. The elucidation of the mechanisms of an increased HERV-W expression in MS could help to develop therapeutic strategies and novel methods for early diagnosis and treatment of the disease.

In this review, we summarized the mechanisms that can affect the expression of *HERV-Wenv* and *ERVWE1* (Table 1), as well as link them to the pathogenesis of MS. We reviewed the epigenetic mechanisms of transcriptional regulation, the role of transcription factors, cytokines, and exogenous viruses. However, it still remains difficult to accurately identify the cause of the elevated expression of the HERV-W genes in MS. The exact role of these retroviral elements in the pathogenesis of MS is still unknown and therefore further research in this area using multi-cohort studies, GWAS, and in vitro approaches will help to establish the contribution of HERVs in MS pathogenesis.

## Figures and Tables

**Figure 1 biology-09-00464-f001:**
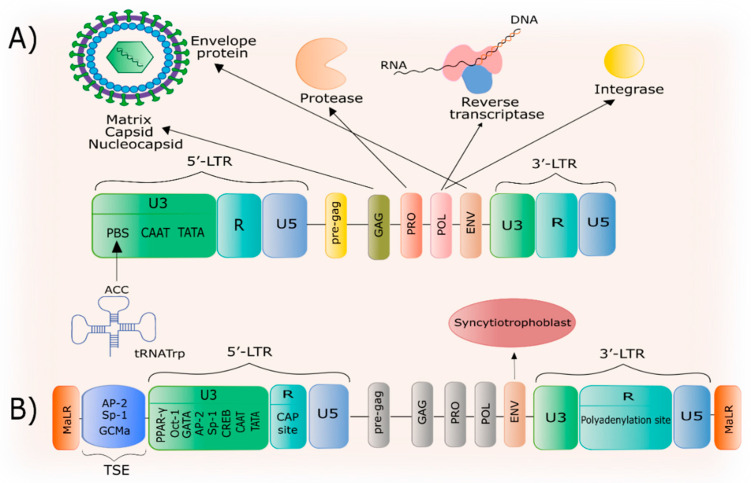
The structure of the HERV-W family and the *ERVWE1* gene. (**A**) The HERV-W family includes four genes, with an up and down stream LTRs. These LTRs consist of U3, R and U5 region. The U3 region of 5′-LTR consist of tRNA^Trp^-binding PBS, CAAT and TATA box sites. The *gag* gene encodes elements of the matrix, capsid and nucleocapsid, while *pro* gene encodes for protease. Viral *pol* gene encodes for reverse transcriptase and integrase. Finally, the envelop protein is coded by the *env* gene. (**B**) The *ERVWE1* gene is “domesticated” and retains the ability to express genes of the HERV-W family. It consists of the *env* gene, called Syncitin-1, with an ORF and is also surrounded by two LTR from 5′ and 3′ ends. Trophoblast-specific enhancer (TSE) containing AP-2, Sp-1 and GCMa binding sites is located upstream of the 5′-LTR. The U3 region of 5′-LTR contains PPAR-γ, Oct-1, GATA, AP-2, Sp-1, CREB, CAAT and TATA box binding sites. R region contains a CAP transcription initiation site at the 5′ end and a polyadenylation site at the 3′ end. The R region of the 3′-LTR includes a polyadenylation site. The whole *ERVWE1* gene is flanked by the MaLR retrotransposon.

**Figure 2 biology-09-00464-f002:**
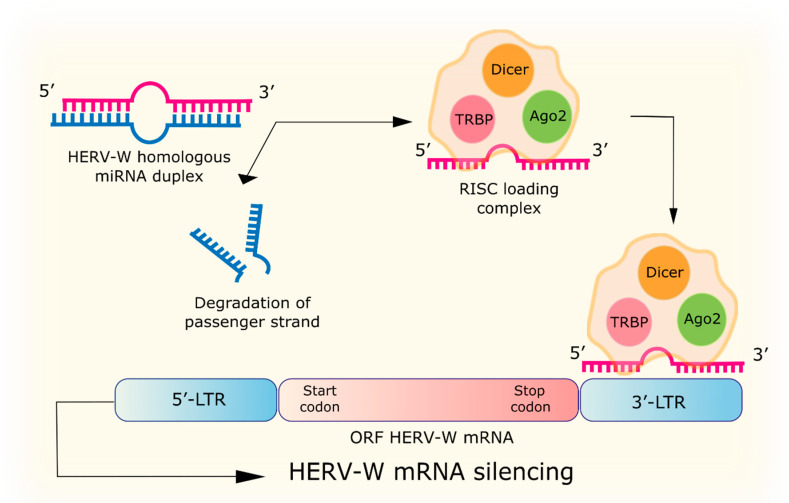
The mechanism of RNA interference. The Dicer enzyme cleaves specific to HERV-W double-stranded miRNA and promotes the loading of each strand into the RISC complex. Dicer binds to the transactivating response RNA-binding protein (TRBP) to facilitate the transfer of dsRNA fragments to Argonaute 2 (Ago2). The least stable 5′ end RNA strand is integrated with the Ago2 protein in RISC, and the second passenger strand is degraded. The RISC complex with guide strand miRNA complementary binds to the 3′-LTR of the mRNA HERV-W and silences it.

**Figure 3 biology-09-00464-f003:**
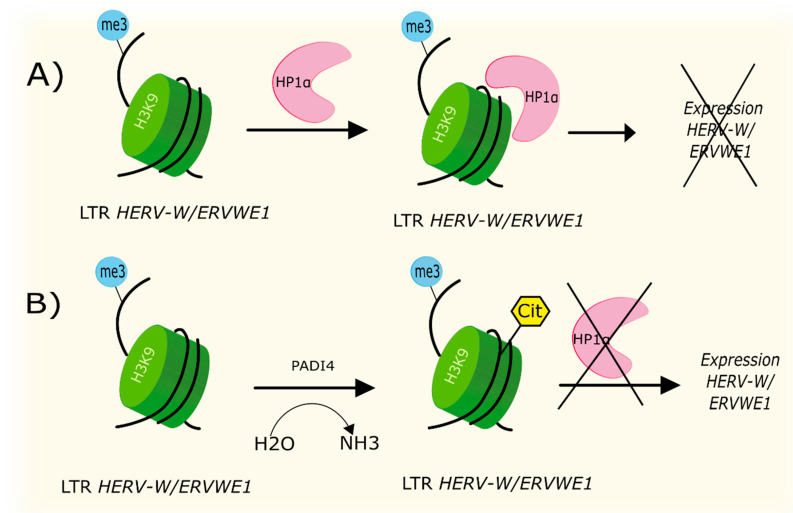
The mechanism of regulation of HERV-W transcription by histone citrullination. (**A**) Repression of HERV-W transcription upon the interaction of HP1α with histone H3K9 in LTR. (**B**) Enhanced expression of HERV-W due to the formation of citrullinated H3K9 histone by PADI4.

**Figure 4 biology-09-00464-f004:**
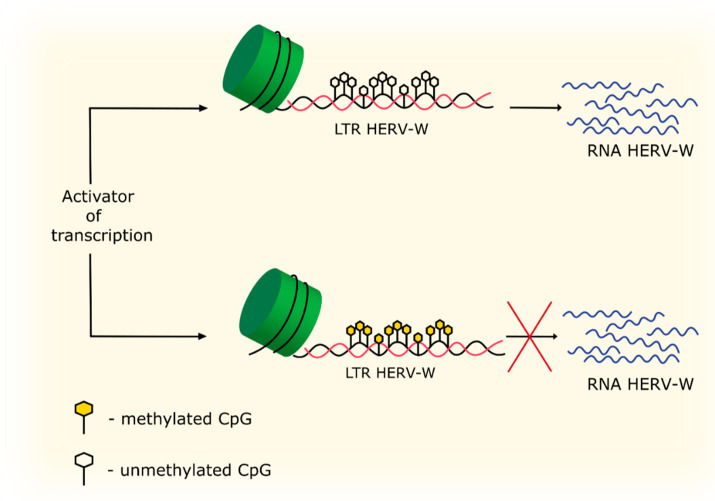
Regulation of HERV-W transcription by methylation of CpG sites in the promoter region.

**Figure 5 biology-09-00464-f005:**
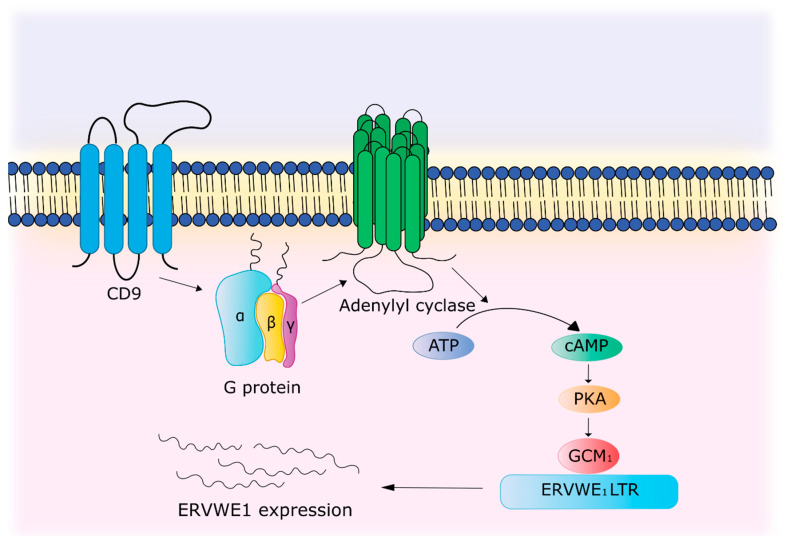
*ERVWE1* expression mechanism due to activation of cAMP/PKA signaling pathway by CD9.

**Table 1 biology-09-00464-t001:** Possible downregulation and upregulation mechanisms HERV-W expression.

Factors	Downregulation Mechanism	References
**RNA interference**	miRNA with homology to HERV-W: hsa-miR-376b, hsa-miR-22, hsa-miR-574, hsa-miR-570, hsa-miR-198, MIMAT0000228, hsa-miR-493-3p, MIMAT0003161, hsa-miR-214, MIMAT0000271, hsa-miR-101, hsa-miR-296, hsa-miR-31, hsa-miR-659, hsa-miR-185, hsa-miR-202, hsa-miR-122a, hsa-miR-24, hsa-miR-506, hsa-miR-632, hsa-miR-376a, hsa-miR-326. siRNA to the p65 gene of the NF-κB subunit reduces URE-LTR activation of the *ERVWE1* promoter by TNFα.	[68,69]
**DNA methylation**	Hypermethylation in the LTR region of HERV-W leads to repression of HERV-W expression.	[83]
	**Upregulation mechanism**	
**Citrullination**	Increased level of PADI4 leads to formation of H3cit8K9me3 histone in promotor region of *ERVWE1.* Local increased citrullination of H3R8 histone localized above H3K9 was associated with increased expression of *HERV-W*/*ERVWE1*.	[70]
**Changes in regulatory region**	The 142T > C mutation promotes the formation of an additional binding site with c-Myb in 3′-LTR of *ERVWE1* and due to this activation of its promoter activity.	[97]
**Activation by immune system**	TNFα and IFN-γ activate the *ERVWE1* promoter. TNFα promotes increased binding of p65 of the NF-κB subunit to a sensitive element located in the enhancer region of the *ERVWE1* promoter. CD9 through cAMP/PKA signaling pathway leads to the activation of GCM1, which binds to the regulatory region of *ERVWE1* and enhances gene expression.	[69,102,103]
**Exogenous viruses**	Tat protein of HIV activate transcription of *MSRV* and *ERVWE1* in in astrocytes through TLR-4 and TNFα. HCMV activate reverse transcriptase necessary for transcription of HERV. ICP0 protein of HSV-1 activate transcription of the *HERV-Wenv* and *MSRVenv* genes. HHV-6A activates MSRVenv expression by interacting with the isoform of CD46-Cyt1 receptor.	[104,106,107,108]

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
