# Peer review of "The Relationship of the Mechanisms of the Pathogenesis of Multiple Sclerosis and the Expression of Endogenous Retroviruses"

_biology, 2020, doi:10.3390/biology9120464_

Round 1
Reviewer 1 Report
Most revisions are valuable, but the presentation of HERV-W in MS with the cited study is a nonsense, for primers and sample conditions could not match required QC criteria:
"In a study by Schmitt 99 et al., expression of multiple human endogenous retroviruses (HERV) type W (HERV-W) Env loci 1 was demonstrated in MS as well as in controls [35] although, an increased expression of several different genes from multiple HERV families in organs and tissues has been documented in patients 102 with neurological disorders. For example, in MS, an increased expression of HERV-H/F Gag protein 103 was measured in monocytes, CD4+ and CD8+ T lymphocytes [36], HERV-H Env and HERV-W Env 104 epitopes on the surface of B cells and monocytes in samples of patients with the active and stable 105 relapsing-remitting form of the disease [37] ".
In face of the many and top-level publications that first discovered this association and further characterized it, this is not acceptable to present such biases in what should be an accurate review. It may be cited, but not here and, to be honest, the study of MS brains at the protein level is much more informative (Van Horssen et al. 2016, MSARD) than multi-organ RT-PCR with any primers and conditions that do not indicate the translation of a pathogenic protein, but may only represent a physiological regulatory role of non-coding HERV RNAs among others, as discussed in other paragraphs.
HERV-W in MS, not to mention earlier papers, was shown with first sequence identification in Perron et Al, PNAS 1997 and then called MSRV, but this was the prototype sequence from which the HERV-W family was unraveled, since not known before: (Blond et al. J. virol. 1999). The detection in MS versus controls with 3 different techniques in two different labs for PCR analyses is shown in Perron et al. Mult. Scler. 2012;18:1721-1736.The latest high-level published papers are certainly: Kremer et al. Ann. Neurol.2013 and PNAS 2019. Reliable PCR primers and conditions for HERV-W in human diseases are now used in MS pateints (Derfuss et al. Mult. Scler.) and are fully described in Supp. mat. of Levet et al. 2016 JCI insights (https://insight.jci.org/articles/view/94387#sd).
So, why introducing with such a bias? Good reviews should rely upon accurate studies and validated protocols. As such, this review is but misleading the readers from the beginning, even if the following is correct.
This is a pity, since the rest of the discussed parts are interesting and provide useful contribution to the debate.
Only introducing the literature on MS and HERV-W is misleading, if needing to cite a special paper, this may be discussed afterwards, but the reference ones must be provided (as in the review from Küry et al. Trends Mol Med. 2018), otherwise most important information is missed and misleaded studies encouraged for no benefit to science and to the patients.
Please upgrade your review, it deserves it.
Author Response
Dear Editor
Thank you for the chance to resubmit the manuscript. We have substantially rewritten it in response to the reviewer’s comments, with detailed responses provided below, we hope that you agree that the manuscript is much improved as a result,
Yours Sincerely,
Ekaterina Martynova
Most revisions are valuable, but the presentation of HERV-W in MS with the cited study is a nonsense, for primers and sample conditions could not match required
QC criteria:
"In a study by Schmitt 99 et al., expression of multiple human endogenous retroviruses (HERV) type W (HERV-W) Env loci 1 was demonstrated in MS as well as in controls [35] although, an increased expression of several different genes from multiple HERV families in organs and tissues has been documented in patients 102 with neurological disorders. For example, in MS, an increased expression of HERV-H/F Gag protein 103 was measured in monocytes, CD4+ and CD8+ T lymphocytes [36], HERV-H Env and HERV-W Env 104 epitopes on the surface of B cells and monocytes in samples of patients with the active and stable 105 relapsing-remitting form of the disease [37] ".
In face of the many and top-level publications that first discovered this association and further characterized it, this is not acceptable to present such biases in what should be an accurate review. It may be cited, but not here and, to be honest, the study of MS brains at the protein level is much more informative (Van Horssen et al. 2016, MSARD) than multi-organ RT-PCR with any primers and conditions that do not indicate the translation of a pathogenic protein, but may only represent a physiological regulatory role of non-coding HERV RNAs among others, as discussed in other paragraphs.
HERV-W in MS, not to mention earlier papers, was shown with first sequence identification in Perron et Al, PNAS 1997 and then called MSRV, but this was the prototype sequence from which the HERV-W family was unraveled, since not known before: (Blond et al. J. virol. 1999). The detection in MS versus controls with 3 different techniques in two different labs for PCR analyses is shown in Perron et al. Mult. Scler. 2012;18:1721-1736.The latest high-level published papers are certainly: Kremer et al. Ann. Neurol.2013 and PNAS 2019. Reliable PCR primers and conditions for HERV-W in human diseases are now used in MS pateints (Derfuss et al. Mult. Scler.) and are fully described in Supp. mat. of Levet et al. 2016 JCI insights (https://insight.jci.org/articles/view/94387#sd).
So, why introducing with such a bias? Good reviews should rely upon accurate studies and validated protocols. As such, this review is but misleading the readers from the beginning, even if the following is correct.
This is a pity, since the rest of the discussed parts are interesting and provide useful contribution to the debate.
Only introducing the literature on MS and HERV-W is misleading, if needing to cite a special paper, this may be discussed afterwards, but the reference ones must be provided (as in the review from Küry et al. Trends Mol Med. 2018), otherwise most important information is missed and misleaded studies encouraged for no benefit to science and to the patients.
Please upgrade your review, it deserves it.
Agree: this part of article was removed and replaced by « Expression of the HERV-W (MS-associated retrovirus env) protein was found in the brains of all studied MS patients during active demyelination at an early stage and during late disease progression [35]. Also, the HERV-Wenv protein was found in the serum of patients with various forms of MS, as well as in infiltrated macrophages, perivascular infiltrates and activated microglial cells [36]. In a study by Kremer et al., it was found that microglial phagocytosis was inhibited by the HERV-Wenv protein, which in turn leads to the accumulation of myelin debris, which prevents neuronal remyelination [37]. Also, this protein reduces the expression of neuroprotective molecules by microglial cells [37]. It’s worth noting that HERV-W Env epitopes was discovered on the surface of B cells and monocytes in samples of patients with the active and stable relapsing-remitting form of the disease [38]. The fact that the HERV-Wenv protein is found in the brain tissues, as well as in the serum and cerebrospinal fluid [39] of patients with MS, suggests that it is in volved in the pathogenesis of the disease and should be carefully studied. » (new line 99-111). References Van Horssen et al. MSARD, 2016, Kremer et al. PNAS, 2019 was cited.
Reviewer 2 Report
The revised manuscript has addressed the concerns making the manuscript more complete especially with respect to pregnancy- since this is a unique physiologic condition which lead to decline in MS symptoms despite ERV present in the placenta.
The table also important since it provides clarity to pathways involved and ERV affected.
Author Response
The revised manuscript has addressed the concerns making the manuscript more complete especially with respect to pregnancy- since this is a unique physiologic condition which lead to decline in MS symptoms despite ERV present in the placenta.
The table also important since it provides clarity to pathways involved and ERV affected.
Agree: We would like to thank the reviewer for this review and pointing out some errors
This manuscript is a resubmission of an earlier submission. The following is a list of the peer review reports and author responses from that submission.
Round 1
Reviewer 1 Report
This review manuscript aims at discussing a very interesting subject. It is about the dysregulation of the human endogenous retrovirus HERV-W-derived loci in multiple sclerosis (MS).
The manuscript tries to summarize the characteristics of human endogenous W retroviruses, and review the putative mechanisms of transcriptional reactivation of the of HERV-Ws, including epigenetic and humoral mechanisms and the impact of exogenous viral infection.
There is a rich literature on the two human endogenous retrovirus-derived loci of the HERV-W family whose products can act as cofactors of multiple sclerosis (MS). The MS-associated retrovirus (MSRV) and ERVWE1. In the human genome there areadditional (over 600) truncated HERV-W copies. There are multiple excellent review articles already published on the subject (including DOI: 10.1080/13550280590952899, 10.1177/1352458517737370, etc). This current review article focuses on the different factors that might be involved in the regulation.
Unfortunately, the review is badly written. The English is very poor. One of the figure legends (Figure 1) is in Russian. The font size changing throughout the manuscript, and some of the references are missing: [Error! Reference source not found., Error! Reference source not found.]. In addition to the formal issues, the author(s) are not familiar with several basic terms. A review article should have a proper overview of the field and could not make such basic mistakes. Furthermore, it should also provide a new angle that has not been discussed before. I am afraid, this work fails to meet the standards. The submitted manuscript gives the impression that an enthusiastic student in the lab wrote a draft that has been never checked by the PI before the submission...
This manuscript is not in the stage that can be published, but even premature for a review process! A work in such a condition should not be sent out from a reputed lab! It is not fair to ask reviewers to spend their precious time on reviewing an article in such a condition!
After reading the pretty good reviews on the subject, I am not convinced about this current one at all.
Factual mistakes:
- Origin of HERV-W:
"The origin of these genetic elements in human DNA is still not known exactly."
It is not correct. We know exactly that ERVs are remnants of exogenous retroviral integrations! None of the copies work as ERVs any longer, they do not reintegrate and generate new copies, but viral particles can be still detected under certain conditions.
- Infectious viral particles:
"The founder member of the HERV-W family is the multiple sclerosis-associated retrovirus (MSRV, 13–14), a presumably complete virus, since it is able to form extracellular, infectious virions."
Not correct! It is not a functional retrovirus, and although viral particles are detectable, no data are convincing enough to show that they can be infectious. The first description of retrovirus-like particles in cell cultures from multiple sclerosis (MS) patients suggested a viral origin of the disease. However, further molecular characterization of this MSRV revealed that it was not a classical infectious exogenous retrovirus. Some chromosomal copies retain potential ORF for retroviral proteins (e.g. env) that have been associated with pathogenesis.
- Section 1.
"Endogenous and exogenous factors affecting the development of MS":
Regarding the part discussing vitamin D deficiency. It is a lengthy discussion about the association of MS and Vitamin D deficiency. However, there is no obvious connection between Vitamin D deficiency and the upregulation of HERV-Ws. Thus, this paragraph seems to do nothing with HERV-W regulation.
- Section 2.
"General characteristics of HERV"
It is meant to be about HERV-W and not in HERV in general...?
- "...Katja Schmitt et.al; proved the presence of promoter activity of incomplete LTR sequences."
Are all of the 654 HERV-W copies expressed in trophoblast?
- Section 3.
"The structure of the HERV-W family"
"The family of endogenous retroviruses HERV-W consists of four genes: gag, pro,pol, env (Fig. 1)."
However, Fig 1 shows a structure of Syncytin-1, expressed from the ERVWE1 locus. Syncytin-1is special, because it represents one of the rare domesticated events of an ERV. Syncytin-1 is a co-opted retroviral envelope that fuses cell membranes during the formation of syncytiotrophoblast in placentation.
As Syncytin-1 is a domesticated gene, having a function in placentation, it cannot be considered as a representative of a HERV-W family of ERVs. For example, the MaLR LTR is not part of the HERV-W at all, it is part of the domesticated regulatory unit, providing its tissue specific expression in the trophoblast. Similar, the trophoblast-specific enhancer (TSE) and TF binding sites are also specific for this locus, etc.
"...Katja Schmitt et.al; proved the presence of promoter activity of incomplete LTR sequences."
Are all of the 654 HERV-W copies expressed in trophoblast?
Section 4.
" Epigenetic mechanisms of regulation of HERV-W expression"
Fig 2. This is a school book figure explaining the mechanism of RNA interference in general.No connection with HERV-W regulation.
Section 4.1
"RNA interference"
The section "RNA interference" presents some conflicting data and not even specific to HERV-W regulation. I can agree only with the conclusion that "It is possible that the mechanism of RNA interference may not directly affect the expression of endogenous retroviruses."
Section 4.2.
"Citrullination"
This section might contain some relevant information, still, references [58-60] have nothing to do with Citrullination.
Section 4.3.
Methylation and acetylation
"It was found that cytokines associated with MS, TNFα and IFN-γ, activate the ERVWE1promoter, which was confirmed by the removal of the domain from the enhancer region of the promoter containing transcription factor binding sites." - no references cited.
"In the work of Weber. S et al. it was shown that the insertion of foreign DNA affects the methylation of CpG sites in various regions of human genes and changes their expression [84]. They later found that transfection of a cell line with the same plasmid pC1–5.6 did not affect the methylation of CpG sites and the level of HERV- W transcription [85]."
Not clear what the message is here.
Section 4.6.
"Activation of HERV-W transcription upon exposure to exogenous viruses"
This aspect has been currently discussed in a review article with the involvement of the PI of this manuscript: doi: 10.3390/v12060643
Minor:
"Endogenous retroviral elements were inherited by humans from his ancestors and appeared in his genome millions of years ago. - very bad English
"The etiology of multiple sclerosis is unknown, but it is known that MS is more common in young women [2-3]. Early diagnosis of the disease is also difficult due to the late manifestation of clinical signs [4]." - Not clear. Seems to be a contradiction.
Author Response
Dear Editor
Thank you for the chance to resubmit the manuscript. We have substantially rewritten it in response to the reviewer’s comments, with detailed responses provided below, we hope that you agree that the manuscript is much improved as a result,
Yours Sincerely,
Ekaterina Martynova
This review manuscript aims at discussing a very interesting subject. It is about the dysregulation of the human endogenous retrovirus HERV-W-derived loci in multiple sclerosis (MS).
The manuscript tries to summarize the characteristics of human endogenous W retroviruses, and review the putative mechanisms of transcriptional reactivation of the of HERV-Ws, including epigenetic and humoral mechanisms and the impact of exogenous viral infection.
There is a rich literature on the two human endogenous retrovirus-derived loci of the HERV-W family whose products can act as cofactors of multiple sclerosis (MS). The MS-associated retrovirus (MSRV) and ERVWE1. In the human genome there areadditional (over 600) truncated HERV-W copies. There are multiple excellent review articles already published on the subject (including DOI: 10.1080/13550280590952899, 10.1177/1352458517737370, etc). This current review article focuses on the different factors that might be involved in the regulation.
Unfortunately, the review is badly written. The English is very poor. One of the figure legends (Figure 1) is in Russian. The font size changing throughout the manuscript, and some of the references are missing: [Error! Reference source not found., Error! Reference source not found.]. In addition to the formal issues, the author(s) are not familiar with several basic terms. A review article should have a proper overview of the field and could not make such basic mistakes. Furthermore, it should also provide a new angle that has not been discussed before. I am afraid, this work fails to meet the standards. The submitted manuscript gives the impression that an enthusiastic student in the lab wrote a draft that has been never checked by the PI before the submission...
This manuscript is not in the stage that can be published, but even premature for a review process! A work in such a condition should not be sent out from a reputed lab! It is not fair to ask reviewers to spend their precious time on reviewing an article in such a condition!
After reading the pretty good reviews on the subject, I am not convinced about this current one at all.
English was edited by Dr Urbanowitch, who also did intellectual contribution to the discussion.
All references was edited.
Factual mistakes:
- Origin of HERV-W:
"The origin of these genetic elements in human DNA is still not known exactly."
It is not correct. We know exactly that ERVs are remnants of exogenous retroviral integrations! None of the copies work as ERVs any longer, they do not reintegrate and generate new copies, but viral particles can be still detected under certain conditions.
Agree: the sentence was changed to “These genetic elements were attained in human DNA as the result of the integration of exogenous retroviral RNA followed the reverse transcription [41] It was suggested that the expression of these viral proteins have had a significant impact on human evolution [32,42].” (new line 109-111)
- Infectious viral particles:
"The founder member of the HERV-W family is the multiple sclerosis-associated retrovirus (MSRV, 13–14), a presumably complete virus, since it is able to form extracellular, infectious virions."
Not correct! It is not a functional retrovirus, and although viral particles are detectable, no data are convincing enough to show that they can be infectious. The first description of retrovirus-like particles in cell cultures from multiple sclerosis (MS) patients suggested a viral origin of the disease. However, further molecular characterization of this MSRV revealed that it was not a classical infectious exogenous retrovirus. Some chromosomal copies retain potential ORF for retroviral proteins (e.g. env) that have been associated with pathogenesis.
Specify your question, please: because this sentence was written by Antonina Dolei in article “MSRV/HERV-W/syncytin and its linkage to multiple sclerosis: The usablity and the hazard of a human endogenous retrovirus”, and published Journal of Neurovirology in 2005.
Section 1.
"Endogenous and exogenous factors affecting the development of MS":
Regarding the part discussing vitamin D deficiency. It is a lengthy discussion about the association of MS and Vitamin D deficiency. However, there is no obvious connection between Vitamin D deficiency and the upregulation of HERV-Ws. Thus, this paragraph seems to do nothing with HERV-W regulation.
Agree: this part of article describes the main possible factors that can affect on the development of multiple sclerosis, including elevated expression of endogenous retroviruses W family. In this case vitamin D is considered as possible risk factor for MS development, but not as directly related to HERV-W expression.
- Section 2.
"General characteristics of HERV"
It is meant to be about HERV-W and not in HERV in general...?
Agree: the title of this part was changed to “General characteristics of HERV-W”
- "...Katja Schmitt et.al., proved the presence of promoter activity of incomplete LTR sequences."
Are all of the 654 HERV-W copies expressed in trophoblast?
Agree: this is a really important question, however, based on data published by Katja Schmitt et.al., we cannot give an exact answer to this question. This study was devoted to studying the expression of the HERV-W loci in multiple sclerosis brain lesions, but not in the trophectoderm.
- Section 3.
"The structure of the HERV-W family"
"The family of endogenous retroviruses HERV-W consists of four genes: gag, pro,pol, env (Fig. 1)."
However, Fig 1 shows a structure of Syncytin-1, expressed from the ERVWE1 locus. Syncytin-1is special, because it represents one of the rare domesticated events of an ERV. Syncytin-1 is a co-opted retroviral envelope that fuses cell membranes during the formation of syncytiotrophoblast in placentation.
As Syncytin-1 is a domesticated gene, having a function in placentation, it cannot be considered as a representative of a HERV-W family of ERVs. For example, the MaLR LTR is not part of the HERV-W at all, it is part of the domesticated regulatory unit, providing its tissue specific expression in the trophoblast. Similar, the trophoblast-specific enhancer (TSE) and TF binding sites are also specific for this locus, etc.
Agree: the structure of ERVWE1 was made separately frow whole HERV-W family
"...Katja Schmitt et.al; proved the presence of promoter activity of incomplete LTR sequences."
Are all of the 654 HERV-W copies expressed in trophoblast?
Agree: this is a really important question, however, based on data published by Katja Schmitt et.al., we cannot give an exact answer to this question. This study was devoted to studying the expression of the HERV-W loci in multiple sclerosis brain lesions, but not in the trophectoderm.
Section 4.
" Epigenetic mechanisms of regulation of HERV-W expression"
Fig 2. This is a school book figure explaining the mechanism of RNA interference in general. No connection with HERV-W regulation.
Agree: at the moment there is no reliable information about the mechanism that starts this process in case of HERV-W expression. In order to make the drawing more specific to the HERV-W, we changed the captions of the main elements to “HERV-W homologous miRNA duplex”, “ORF HERV-W mRNA”, “HERV-W mRNA silencing” and slightly clarified description under it.
Section 4.1
"RNA interference"
The section "RNA interference" presents some conflicting data and not even specific to HERV-W regulation. I can agree only with the conclusion that "It is possible that the mechanism of RNA interference may not directly affect the expression of endogenous retroviruses."
Agree: the sentence was added (new line 187-188) “Currently, there is contradictory data about miRNA effects on development of MS and HERV-W expression.” Under this paragraph, we suggested a possible reason for the conflicting data obtained by researchers from different groups: “These conflicting results could be explained by the fact that each study used different patient populations without considering the relapse phase of disease.” (new line 195-196)
Section 4.2.
"Citrullination"
This section might contain some relevant information, still, references [58-60] have nothing to do with Citrullination.
Agree: sentences “In MS, a decreased expression of the EHMT2 gene encoding the histone H3 methyltransferase has been demonstrated [83]. Interestingly, methylation of this gene has been shown to be necessary for repression of HERV-W transcription [68]. An increased EHMT2 expression is also found in the late stages of oligodendrocyte differentiation, and shown to contribute to H3K9 methylation and chromatin compaction necessary for differentiation of oligodendrocyte progenitor cells (OPCs) [84], which is impaired in patients with MS [85].” as second paragraph (new line 251-256).
Section 4.3.
Methylation and acetylation
"It was found that cytokines associated with MS, TNFα and IFN-γ, activate the ERVWE1promoter, which was confirmed by the removal of the domain from the enhancer region of the promoter containing transcription factor binding sites." - no references cited.
Agree: reference was added “It was found that cytokines often associated with MS, such as tumor necrosis factor α (TNFα) and interferon-γ (IFN-γ), activate the ERVWE1 promoter [67].” (new line 298-299)
"In the work of Weber. S et al. it was shown that the insertion of foreign DNA affects the methylation of CpG sites in various regions of human genes and changes their expression [84]. They later found that transfection of a cell line with the same plasmid pC1–5.6 did not affect the methylation of CpG sites and the level of HERV- W transcription [85]."
Not clear what the message is here.
Agree: there was added sentence “These results suggest that exogenous retroviral elements are not affecting the expression of endogenous retroviruses by using the CpG site methylation mechanism.” (new line 266-267)
Section 4.6.
"Activation of HERV-W transcription upon exposure to exogenous viruses"
This aspect has been currently discussed in a review article with the involvement of the PI of this manuscript: doi: 10.3390/v12060643
Almost agree: yes, at this article we reveal more detailed mechanisms by which exogenous viruses can influence the expression of endogenous retroviruses. For example, Tat protein of HIV can modulate expression of MSRV and ERVWE1 in different tissue, or ICP-0 protein of HSV-1 activate transcription endogenous retroviruses in vitro.
Minor:
"Endogenous retroviral elements were inherited by humans from his ancestors and appeared in his genome millions of years ago. - very bad English
Agree: sentence was edited “Endogenous retroviral elements are believed to have integrated in our ancestors’ DNA millions of years ago.” (new line 23-24)
"The etiology of multiple sclerosis is unknown, but it is known that MS is more common in young women [2-3]. Early diagnosis of the disease is also difficult due to the late manifestation of clinical signs [4]." - Not clear. Seems to be a contradiction.
Agree: sentence was edited to “There are multiple hypotheses proposed to establish the etiology of MS, showing the multifactorial nature of the disease. There are several risk factors associated with the disease including gender (often diagnosed in women) and age (most common in young age) of the patient [2,3].” (new line 38-41)

Reviewer 2 Report
This review from Lezhnyova et al. represents an interesting and useful purpose in the domain.
It should contribute to a better understanding of many molecular mechanisms and pathways underlying MS pathogenesis and HERV expression in disease. The authors particularly shed light on molecular features affecting DNA expression, RNA transcription and post-translational modifications of proteins that are not commonly discussed in face of MS pathogenesis, though more often when dealing with HERVs.
However, this manuscript presents a succession of analyses and of their corresponding scientific background without consistent global picture and synthetic discussion, to make it the valuable contribution expected from the title (and from author’ s intention, I believe).
Thus, better structured and presented analyses in a global frame leading to a synthetic overview of these aspects should be provided in a revised version, which I would encourage to make.
Detailed comments on the text:
-References of journals are weird and inconsistent between themselves and with usual abbreviations. Even if an unexpected format used by MDPI, they are insconsitent between them.
In addition some cited references are said to be missing within the text itself. e.g.: PJAon for Annals of neurology (Ann. Neurol.) and PNJoI for Journal of Immunology (J. Immunol.), etc.
-Inaccurate statements in the introduction:
° the latitude/sun exposure/Vit. D relationship with MS is not recent at all. It was re-actualized but has been discussed in early times of MS epidemiology. Just search for early papers.
Moreover, it is linked to a risk, not a “cause” of MS (lines 37-38 present diverging terminology).
°line 43: “sites found close to genes.. with different expression” sounds like overinterpretation of genomic studies that do not really provide evidence for an involvement of Vit. D, or this should be explained differently.
°line 65 and 67: cited references are not very accurate ones, all the more for ALs and HERV-K. A good review of HERVs and neurological diseases may better be cited (Kury et al., 2018). Ref 2’ cited for “other diseases” applies to MS, and should be cited in line 65.
° ref 25 described High-throughput sequencing now known not to be appropriate for repeated elements. Conclusions may therefore not be reliable.
°Line 78: “the origin of...(HERVs) is still not known exactly”. Yes it is: germline infection by exogenous retroviruses and transmission to offspring, also explained and illustrated in (Kury et al., 2018).
° Line 110-111: the SU and TM domains are part of the retroviral envelope protein. The envelope protein is not different from the transmembrane domain, it comprises it before furin cleavage and maintained assembly by disulfide bonds. Signal peptide + SU +TM are encoded in the env orf and encode the precursor ENV glycoprotein.
°Line 120: the ref 47 is about HERV-K, not HERV-W.
°figure 1: the title was left in Russian.
°Figure 4: the presented case if CpG methylation on LTRs (non coding -normally), which is the reverse situation of what is explained for coding regions in the text. This should be made clearer to the reader, as whether hypomethylation is favouring HERV expression or not cannot be well understood from the figure and the text, as they are presented and explained.
° Lines 28 Lines 295-298: the effects of HERV-W on BBB is interesting, but here lacks a recent and important reference to a very accurate study (Duperray et al., 2015).
°Line 314: ICP0 and not “ISP0”
Cited references in comments:
Kury, P., Nath, A., Creange, A., Dolei, A., Marche, P., Gold, J., Giovannoni, G., Hartung, H.P., and Perron, H. (2018). Human Endogenous Retroviruses in Neurological Diseases. Trends Mol Med 24, 379-394.
Author Response
Dear Editor,
Thank you for the chance to resubmit the manuscript. We have substantially rewritten it in response to the reviewer’s comments, with detailed responses provided below, we hope that you agree that the manuscript is much improved as a result,
Yours Sincerely,
Ekaterina Martynova
This review from Lezhnyova et al. represents an interesting and useful purpose in the domain.
It should contribute to a better understanding of many molecular mechanisms and pathways underlying MS pathogenesis and HERV expression in disease. The authors particularly shed light on molecular features affecting DNA expression, RNA transcription and post-translational modifications of proteins that are not commonly discussed in face of MS pathogenesis, though more often when dealing with HERVs.
However, this manuscript presents a succession of analyses and of their corresponding scientific background without consistent global picture and synthetic discussion, to make it the valuable contribution expected from the title (and from author’ s intention, I believe).
Thus, better structured and presented analyses in a global frame leading to a synthetic overview of these aspects should be provided in a revised version, which I would encourage to make.
Detailed comments on the text:
-References of journals are weird and inconsistent between themselves and with usual abbreviations. Even if an unexpected format used by MDPI, they are insconsitent between them.
In addition some cited references are said to be missing within the text itself. e.g.: PJAon for Annals of neurology (Ann. Neurol.) and PNJoI for Journal of Immunology (J. Immunol.), etc.
English was edited by Dr Urbanowitch, who also did intellectual contribution to the discussion.
All references was edited.
-Inaccurate statements in the introduction:
° the latitude/sun exposure/Vit. D relationship with MS is not recent at all. It was re-actualized but has been discussed in early times of MS epidemiology. Just search for early papers.
Agree: sentence was edited “Numerous studies have shown a correlation between a decrease in the level of ultraviolet radiation with an increase of the latitude of the region of residence and an increase in the risk of MS and relapse of disease [9-11].” (new line 51-53)
Moreover, it is linked to a risk, not a “cause” of MS (lines 37-38 present diverging terminology).
Agree: the “a possible cause of the illness is vitamin D” was edited to “potential endogenous risk factor of the illness is vitamin D deficiency” (new line 49)
°line 43: “sites found close to genes.. with different expression” sounds like overinterpretation of genomic studies that do not really provide evidence for an involvement of Vit. D, or this should be explained differently.
Agree: sentence was edited “The role of vitamin D in MS was further confirmed by Ramagopalan et al as an increased binding of vitamin D receptor (VDR) was found on genes associated with risk of MS [12].” (new line 53-55)
°line 65 and 67: cited references are not very accurate ones, all the more for ALs and HERV-K. A good review of HERVs and neurological diseases may better be cited (Kury et al., 2018). Ref 2’ cited for “other diseases” applies to MS, and should be cited in line 65.
Agree: the reference was edited “Many studies have shown the relationship between various neurodegenerative diseases (MS, amyotrophic lateral sclerosis, Alzheimer's disease), and expression of endogenous retroviruses genes [31,32].” (new line 95-97)
° ref 25 described High-throughput sequencing now known not to be appropriate for repeated elements. Conclusions may therefore not be reliable.
Almost agree: High-throughput sequencing not suitable for short repeated elements like trinucleotide repeats, hexanucleotide repeat and etc. But in this case HERV belong to the long terminal repeat retrotransposons repetitive element class. In this study Katja Schmitt et.al. performed sequencing of 280 and 330 bp transcript regions after RT-PCR 5' and 3' regions of the HERV-W family env gene using 454 large scale pyrosequencing technology (Roche GS-FLX 454) with an average read length from 330 bp to 500 bp. So, we believe, in this case, this type of sequencing can be used in relation to HERV-W transcripts.
°Line 78: “the origin of...(HERVs) is still not known exactly”. Yes it is: germline infection by exogenous retroviruses and transmission to offspring, also explained and illustrated in (Kury et al., 2018).
Agree: the sentences were edited to “These genetic elements were attained in human DNA as the result of the integration of exogenous retroviral RNA followed the reverse transcription [41]. It was suggested that the expression of these viral proteins have had a significant impact on human evolution [32,42].” and reference to Kury et al., 2018 was added (new line 110-111).
° Line 110-111: the SU and TM domains are part of the retroviral envelope protein. The envelope protein is not different from the transmembrane domain, it comprises it before furin cleavage and maintained assembly by disulfide bonds. Signal peptide + SU +TM are encoded in the env orf and encode the precursor ENV glycoprotein.
Agree: was edited from “the env gene encodes transmembrane elements and the envelope of the virus.” to “the env gene encodes the precursor of the envelope glycoprotein.” (new line 139-140)
°Line 120: the ref 47 is about HERV-K, not HERV-W.
Agree: reference [47] was deleted
°figure 1: the title was left in Russian.
Agree: the title of Fig.1. was translated in English
°Figure 4: the presented case if CpG methylation on LTRs (non coding -normally), which is the reverse situation of what is explained for coding regions in the text. This should be made clearer to the reader, as whether hypomethylation is favouring HERV expression or not cannot be well understood from the figure and the text, as they are presented and explained.
Agree: the sentence was added “Thus, hypermethylation in the LTR region of HERV-W can potentially inhibit the expression of these genes.” (new line 247-248)
° Lines 28 Lines 295-298: the effects of HERV-W on BBB is interesting, but here lacks a recent and important reference to a very accurate study (Duperray et al., 2015).
Almost agree: at this part is described the potential relationship between increased BBB permeability in MS and elevated HERV-W expression, but not the effect of HERV-W on permeability of BBB. I suppose, that information from Duperray et al., 2015 will be suitable in chapter 2. “General characteristics of HERV-W”, therefore the sentence “In an in vitro study, the expression of the MS-associated retrovirus (MRSV) recombinant protein in the brain microvascular endothelial cells increased the expression of ICAM-1, which is associated with the interaction of the MSRV protein with TLR-4 receptors on the surface of endothelial cells [50].” was added (new line 125-128)
°Line 314: ICP0 and not “ISP0”
Agree: “ISP0” was edited to “ICP0” (new line 326)
Cited references in comments:
Kury, P., Nath, A., Creange, A., Dolei, A., Marche, P., Gold, J., Giovannoni, G., Hartung, H.P., and Perron, H. (2018). Human Endogenous Retroviruses in Neurological Diseases. Trends Mol Med 24, 379-394.
Agree:We added this reference to the manuscript

Reviewer 3 Report
MS etiology despite the clinical importance remains largely unknown.
By the time it is diagnosed the disease is already advanced.
When describing in the introduction the possible etiologic causes involvement of bacterial induced altered immune response ie. Smegmatis is not mentioned.
The other aspect that could be highlighted is presence of retroviruses in the placenta, and at the same time when patients are pregnant MS undergo regression of symptoms- as it does for other autoimmune diseases.
On the other hand altered expression of retrovirus can cause pregnancy pathologies such a preeclampsia.
Also retroviruses could reach the maternal system since trophoblastic cells circulate and even remain in the body for some time and even for many years- when they may flare up scleroderma.
However, protection is pregnancy dependent since after delivery there is a flare up of MS.- An apparent paradox?
A Table would help clarify what compounds can upregulate/downregulate the virus expression- since it is usually silent and is not translated. The text mentions but it is complex reading.
The pathways for retrovirus activation/silencing is adequate. The first figure legend is in Russian needs to provide more detail.
In some areas it shows error with references - needs to correct.
Author Response
Dear Editor,
Thank you for the chance to resubmit the manuscript. We have substantially rewritten it in response to the reviewer’s comments, with detailed responses provided below, we hope that you agree that the manuscript is much improved as a result,
Yours Sincerely,
Ekaterina Martynova
English was edited by Dr Urbanowitch, who also did intellectual contribution to the discussion.
All references was edited.
MS etiology despite the clinical importance remains largely unknown.
By the time it is diagnosed the disease is already advanced.
When describing in the introduction the possible etiologic causes involvement of bacterial induced altered immune response ie. Smegmatis is not mentioned.
Agree: the paragraphs about bacterial component as possible etiological cause was added (new line 83-94). We know that M.Smegmatis, which expresses an encephalitogenic chimeric protein, is used as a trigger factor in a mouse model of a relapsing experimental autoimmune encephalomyelitis [1) doi:10.18632/oncotarget.15662, 2) doi: 10.4049/jimmunol.0804263]. Could you please clarify what information about M.Smegmatis we should discuss in this part of the review?
The other aspect that could be highlighted is presence of retroviruses in the placenta, and at the same time when patients are pregnant MS undergo regression of symptoms- as it does for other autoimmune diseases.
Agree: sentences were added “Hormone levels have also been implicated in late onset of the disease, as the first episode of MS neurological symptoms is often diagnosed in older women who have been pregnant as compared to those who have not [4,5].” (new line 41-43). For underlying expression HERV-W during pregnancy, sentence was edited to “The expression of endogenous retroviruses can be detected not only in healthy people, for example during the pregnancy [33], but also in various pathologies [34].”
On the other hand, altered expression of retrovirus can cause pregnancy pathologies such a preeclampsia.
Agree: Preeclampsia is one of the most relevant problems of modern obstetrics, associated with high maternal and perinatal mortality. Experiments have shown insufficient cell fusion in trophoblast cells in PE patients compared with controls. A decreased level of expression of the protein Synticin-1 and Synticin-2 was also found [doi:10.1177/1933719111404608]. We believe that studying the mechanisms of regulation of the expression of genes of the HERV-W family will help scientists to shed light on the pathogenesis of both preeclampsia and multiple sclerosis.
Also retroviruses could reach the maternal system since trophoblastic cells circulate and even remain in the body for some time and even for many years- when they may flare up scleroderma.
Agree: scleroderma is an autoimmune disease characterized by increased collagen synthesis, which leads to tissue sclerosis. However, in the case of endogenous human retroviruses of the W family, a decreased level of env gene expression was found in patients with the active form of morphea [doi:10.5114/ada.2017.65621].
However, protection is pregnancy dependent since after delivery there is a flare up of MS.- An apparent paradox?
Agree: at the moment the exact reason for the increase in the frequency of relapses is unknown. There are studies describing a possible mechanism of remission due to an increased level of estrogen [doi:10.1177/1352458506071171], the level of which reaches its maximum values during the third trimester of pregnancy, and at the same time the frequency of relapses is reduced to a minimum [doi:10.1093/brain/awh152]. Thus, the study of this issue is of great importance for understanding the pathogenesis of MS, as well as for the development of alternative treatments.
Table would help clarify what compounds can upregulate/downregulate the virus expression- since it is usually silent and is not translated. The text mentions but it is complex reading.
Agree: the table was added in section “Conclusion”
The pathways for retrovirus activation/silencing is adequate. The first figure legend is in Russian needs to provide more detail.
Agree: Fig.1. was edited and legend was added
In some areas it shows error with references - needs to correct.
Agree: all references were checked and edited
